# Quinolones: Mechanism, Lethality and Their Contributions to Antibiotic Resistance

**DOI:** 10.3390/molecules25235662

**Published:** 2020-12-01

**Authors:** Natassja G. Bush, Isabel Diez-Santos, Lauren R. Abbott, Anthony Maxwell

**Affiliations:** Department of Biological Chemistry, John Innes Centre, Norwich Research Park, Norwich NR4 7UH, UK; tash.bush@jic.ac.uk (N.G.B.); Isabel.Diez-Santos@jic.ac.uk (I.D.-S.); lra10@leicester.ac.uk (L.R.A.)

**Keywords:** fluoroquinolones, DNA gyrase, topoisomerases, antibacterials, DNA topology, supercoiling, antibiotic resistance

## Abstract

Fluoroquinolones (FQs) are arguably among the most successful antibiotics of recent times. They have enjoyed over 30 years of clinical usage and become essential tools in the armoury of clinical treatments. FQs target the bacterial enzymes DNA gyrase and DNA topoisomerase IV, where they stabilise a covalent enzyme-DNA complex in which the DNA is cleaved in both strands. This leads to cell death and turns out to be a very effective way of killing bacteria. However, resistance to FQs is increasingly problematic, and alternative compounds are urgently needed. Here, we review the mechanisms of action of FQs and discuss the potential pathways leading to cell death. We also discuss quinolone resistance and how quinolone treatment can lead to resistance to non-quinolone antibiotics.

## 1. Introduction

The quinolone antibiotics (Figure 1) are the most successful class of topoisomerase inhibitors to date. They are synthetic antimicrobials with the initial compound, nalidixic acid, being discovered as a by-product of chloroquine synthesis in 1962 [1,2]. They are used to treat bacterial infections caused by both Gram-positive and Gram-negative bacteria, including, but not limited to, urinary tract infections (UTIs), pyelonephritis, gastroenteritis, sexually-transmitted diseases, such as Gonorrhoea, tuberculosis [3], prostatitis, community-acquired pneumonia and skin and soft-tissue infections [4,5]. However, due to an increase in resistance and issues surrounding toxicity, their use in the treatment of mild infections has been contraindicated [6]. The global rise in antibiotic resistance has galvanised research into new antibiotics against both well-established targets and new targets. It has also sparked further research into antibiotics whose mode of killing is less well-established, as well as how bacteria become resistant to them. This is certainly true in the case of quinolones. In this review, the current knowledge on the mode of action of quinolones, how they kill bacteria and known pathways to resistance will be discussed. Moreover, we will review the current literature on sublethal quinolone exposure leading to resistance to quinolone and non-quinolone antibiotics.

## 2. Background on Quinolones

The discovery of nalidixic acid (Figure 1) was reported in 1962 during the analogue synthesis of a lead structure: 7-chloro-1-ethyl-1,4-dihydro-4-oxo-3-quinolinecarboxylic acid, which was detected as an impurity in the synthesis of chloroquine (an antimalarial agent). From the analogues produced, nalidixic acid was notable due to its moderate antibacterial activity against Gram-negative species (except against *Pseudomonas aeruginosa*), including *Escherichia coli*, both in vitro and in vivo [1]. Several years later, nalidixic acid was released for clinical use for the treatment of uncomplicated UTIs [7]. This sparked the synthesis of additional analogues, although with little improvement over nalidixic acid in terms of spectrum of activity and serum concentration [1,8,9]. Other analogues included oxolinic acid, which was also introduced to the clinic, and these compounds, along with nalidixic acid (although, in relation to its structure, nalidixic acid is a 1,8 naphthyridone and not a true quinolone [1,2]), are considered first-generation quinolones [4,10,11]. There are many proposals concerning how the generations of quinolones should be defined. Of particular note, there is the suggestion of quinolone generations characterised by their structure, mechanism and their killing pathway [12], and there is the classification by structure, in vitro activity and clinical use [13]. In this review, we are referring to the quinolone generations classified by their clinical uses and spectrum of antibacterial activity outlined by Andriole [11]. Continued optimisation of the quinolones led to a fluorine atom being substituted onto carbon 6 (C-6) of the quinolone scaffold (Figure 2), producing a fluoroquinolone (FQ). The first fluoroquinolone was Flumequine, which, after brief use in the clinic, was abandoned due to ocular toxicity [13,14]. Another key modification that enhanced potency was the incorporation of a piperazine ring onto C-7 [2,15,16,17,18]. This C-7 addition, along with the C-6 fluorine, formed the second-generation quinolones, which have a broader scope of activity and better bioavailability, as well as improved pharmacokinetic and pharmacodynamic properties [8,16,19,20,21]. They were also less toxic and were less susceptible to single point mutations that led to high levels of resistance seen against the first-generation quinolones [7,8,17,22]. The second-generation quinolone class began with norfloxacin (Figure 1) [15], which proved to be effective in the treatment of genitourinary and gastrointestinal tract infections, as well as increased activity against *P. aeruginosa* [15,16,21,22,23]. However, it was ciprofloxacin (Figure 1) that was the first quinolone that showed effective systemic activity [8,17,24,25]. Ciprofloxacin is listed as a first-line treatment for low-risk febrile neutropenia within cancer patients and a second-line treatment for cholera, as well as being employed clinically against a range of UTIs, such as those caused by *Pseudomonas aeruginosa* [26]. It has also been demonstrated to be effective in the treatment of Enterobacteriaceae-induced osteomyelitis, prostatitis and septicaemia [8].

Following the success of ciprofloxacin, the observed structure-activity relationships (SARs) were explored further (Figure 2). This medicinal chemistry effort produced a wide range of newer-generation FQs (third and fourth generations) that have even broader spectra of activity, greater efficacy and a lower prevalence of resistance [27]. Sparfloxacin and moxifloxacin (Figure 1) are the better-known compounds of the third and fourth generations, respectively, and are amongst the first quinolones to show significant potency against Gram-positive bacteria [11]. Furthermore, *Mycobacterium tuberculosis*, the bacterial species that causes tuberculosis (TB; currently the world’s deadliest bacterial infectious disease to date), is susceptible to the FQs, and both moxifloxacin and levofloxacin (a second-generation quinolone) have been used in the treatment of multidrug-resistant (MDR) infections [3,28]. Despite their success, some promising FQs, such as trovafloxacin and grepafloxacin (Figure 3), have had to be withdrawn from the clinic due to safety concerns [8,29,30]. However, many have remained in the clinic, with ciprofloxacin continuing to be one of the most clinically important antibiotics to date. In fact, the World Health Organisation has categorised ciprofloxacin (amongst other FQs) as a critically important antibiotic [31].

A major reason for the comparative success of FQs is that they target the bacterial type II topoisomerases, DNA gyrase (gyrase) and DNA topoisomerase IV (topo IV) [33,34,35]. DNA topoisomerases are enzymes that catalyse the interconversion of different topological forms of DNA (e.g., relaxed-supercoiled and catenated-decatenated) and are crucial for several DNA-associated processes, such as replication and transcription [36]. All topoisomerases can relax DNA, but only gyrase can introduce negative supercoiling [37,38,39]. Gyrase is present and essential in all bacteria but absent from higher eukaryotes (e.g., humans), making it an ideal target for antibacterials; however, gyrase does occur in plants and plasmodial parasites [40,41]. Eukaryotes possess a related enzyme, DNA topoisomerase II (topo II), but it is sufficiently different from bacterial gyrase and topo IV such that these enzymes can be selectively targeted. Gyrase and topo IV are both heterotetramers, consisting of GyrA and GyrB (A_2_B_2_) in the case of gyrase and ParC and ParE (C_2_E_2_) in the case of topo IV [35]. Through extensive structural and mechanistic studies, the mechanisms of action of gyrase (Figure 4) and topo IV are well-understood [35,42]. As with all type II topoisomerases, the mechanism entails the binding of two segments of DNA, a G (or Gate) segment and a T (or Transported) segment. The enzyme cleaves the G segment in both strands of the DNA, leaving a four-base stagger, involving amino acid residues in both subunits and entailing the formation of covalent bonds between the 5′ ends of the broken DNA and the active site tyrosines in GyrA or ParC [35,42]. This allows passage of the T segment through the G-segment break (strand passage), enabling the alterations in DNA topology, such as relaxation or decatenation. In the case of gyrase, the T and G segments are located close together on the same piece of DNA, enabling vectorial strand passage and, thus, the introduction of negative supercoils. The mechanism requires the hydrolysis of two ATP molecules [43,44], although the exact timing and role of the ATP is yet to be determined.

The quinolones inhibit DNA supercoiling and relaxation by binding to both gyrase and DNA and stabilising the gyrase-DNA-cleaved complex [33,34]. This is also true for topo IV, which is the primary target in a number of Gram-positive species [45,46]. However, this is often dependent on the specific quinolone, and some quinolones have been shown to target both enzymes equally [5,7,47,48,49]. A thorough review by Cheng et al. [50] provides more information about the differential targeting of gyrase and topo IV by quinolones and the consequences thereof.

The quinolones have been shown to have interactions with both subunits of the enzyme (GyrA and GyrB for gyrase and ParC and ParE for topo IV). Research into the nature of FQ binding has led to several potential mechanisms and the suggestion that it may involve several steps [51,52]. Crystal structures published in 2009 and 2010 indicated a convincing model that is likely to represent the principle and most stable mode of binding [53,54,55]. In this model, the drug is seen to be intercalated between DNA bases at the DNA-cleavage site (Figure 5); intercalation of quinolones into DNA were proposed from earlier works [56,57,58]. The intercalation model represents a now well-established explanation for FQ binding, in which the drug binds at the DNA gate region of the enzyme whilst partially intercalating into the substrate DNA (Figure 5). It appears that the drugs may take advantage of the modified DNA structure at the cleavage site to intercalate between DNA bases at the sites of breakage. Within X-ray crystal structures, one FQ molecule has been found to intercalate between the bases at each DNA break, inducing a kink. The C-7 substituent of the FQ protrudes out of the DNA slightly, avoiding unfavourable clashes with the DNA bases on either side. This model also explains the increased action of the FQs over the first-generation quinolones, where the fluorine substituent likely perturbs the electronic balance of the partially aromatic structure and strengthens pi-stacking interactions with the DNA bases [54]. The carbonyl substituents at C-3 and -4 contribute essential contacts and underpin the formation of a water-metal ion bridge. This water-metal ion bridge was found to mediate the interactions between the drug and the target enzyme; this consists of a noncatalytic Mg^2+^ ion in an octahedral complex with four water molecules and the FQ C-3/C-4 carbonyl oxygens. Two of these water ligands interact with enzyme residues, S83 and D87, in GyrA (using *E. coli* numbering), completing the bridge. Interactions between position 466 in GyrB and the C-7 ring of the FQ are also important for binding of the compound [52,59,60,61]. These interactions between the quinolones and the topoisomerase-DNA complex trap the topoisomerase on the DNA, making the enzyme unable to supercoil or relax the DNA. This topoisomerase-DNA-quinolone complex also transforms the enzyme into a poisonous protein that blocks the replication [62,63] and transcription machinery [64], potentially causing lethal double-strand breaks [45,46,64,65,66].

## 3. Quinolone Lethality

Quinolones kill bacteria slowly or quickly, depending on their concentrations. At concentrations that are twice the minimum inhibitory concentration (MIC) value, bacteria are killed after an overnight quinolone treatment; whereas, at concentrations 5–10 times the MIC value, bacteria die after a few hours of quinolone exposure [67,68]. How slow or rapid killing happens is not fully understood. We know that the first stage in quinolone lethality is the binding of the quinolone to a topoisomerase-DNA cleavage complex. Cleavage complexes contain broken DNA that cannot be resealed by the same topoisomerase if the quinolone is present. However, cleavage complexes and their “hidden” DNA breaks are reversible, so there must be additional events that cause bacterial death [65]. Depending on whether the cleavage complex is processed, it is thought that bacterial death can arise in two different ways. If the cleavage complex is not processed, DNA replication and transcription are blocked, eventually leading to cell death: slow death. If the cleavage complex is processed (either by removing the gyrase from the DNA with an unknown protein or because the gyrase subunits dissociate) and the broken DNA is not repaired, it causes chromosome fragmentation, which quickly kills the cell: rapid death. The presence of broken DNA and, perhaps, cleavage complexes also cause the accumulation of intracellular ROS (reactive oxygen species), which can lead to more DNA breaks (Figure 6) [50,69]. The quinolone-induced DNA damage can be repaired (at least in part), which, as we will mention later, can have important consequences for the survival of the cell to quinolone and non-quinolones antibiotics.

### 3.1. Slow Death

#### 3.1.1. Block of Replication and Transcription

At low concentrations, quinolones block replication and transcription by inhibiting gyrase and topo IV, which are essential enzymes during replication and transcription [48,70]. Replication fork progression causes positive supercoils to build up ahead of the fork [71,72]. These positive supercoils need to be resolved, as a build-up will cause a large amount of torsional stress [73] that can stall replication. To relieve the torsional stress, the replication fork may rotate causing the development of precatenanes behind the fork [71,74,75,76]. These precatenanes can become tangled and knotted if left unresolved, leading to incomplete segregation at the end of replication [71,76]. Gyrase acts ahead of the replication fork, removing positive supercoils so that replication and transcription can continue unhindered. Topo IV, on the other hand, works behind the replication fork, unlinking precatenanes that could prevent cell division [77]. The main components of the replication and transcription machinery, the DNA and RNA polymerases, are blocked by gyrase-quinolone-DNA complexes [63,64], and the same happens to replication forks [78]. However, stopping replication had little effect on the lethal activity of the quinolones [79], and it is therefore unlikely to be the cause of quinolone-induced death. In fact, quinolone-induced death correlated with the release of DNA breaks from gyrase-cleavage complexes [65]. These lethal DNA breaks do not come from the blockage of replication [80,81] but happen after the trapped gyrase is removed from the DNA.

In contrast to quinolones, drugs that stabilise eukaryotic topoisomerase-DNA complexes can generate lethal DNA breaks when replication forks collide with cleavage complexes and the trapped topoisomerase has not been removed from the DNA [82,83]. Camptothecin, a stabiliser of the eukaryotic topoisomerase I-DNA cleavage complex, inhibits DNA replication and generates DNA breaks [82,84]. It also causes the formation of double-strand breaks when the replication fork collides with the cleavage complex [83,85]. A similar situation might happen with topo II-DNA cleavage complexes, as m-AMSA, an inhibitor of topo II, is less lethal in the presence of a DNA synthesis inhibitor, which indicates that its lethality depends on the replication of DNA [86].

#### 3.1.2. Inhibition of DNA and RNA Synthesis

The main consequence of the arrest of replication forks and transcription bubbles is the inhibition of DNA and RNA synthesis. DNA and RNA synthesis rates quickly decrease in the presence of quinolones [62,65,87], and this correlates with the inhibition of growth [87,88]. However, the quinolone-induced inhibition of DNA synthesis is reversible—that is, DNA synthesis resumes upon the removal of the drug, so, like inhibition of replication, it is unlikely to cause cell death [89]. Nevertheless, it has been proposed that quinolone slow killing (which happens when bacteria are given long quinolone treatments at twice the MIC) might be caused by secondary events stimulated by the inhibition of replication [51].

### 3.2. Rapid Death

#### 3.2.1. Processing of the Quinolone-Poisoned Gyrase

As mentioned before, quinolones can quickly kill bacterial cells at concentrations over the MIC, and this lethality mostly appears when the quinolone-poisoned gyrase subunits disassociate or are removed from the DNA. There are three ways in which a poisoned topoisomerase can be removed from the DNA that differ in the need for protein synthesis and aerobic conditions [90]. First-generation quinolones, such as oxolinic or nalidixic acid, are not lethal in the presence of a protein-synthesis inhibitor (e.g., chloramphenicol) or under anaerobic conditions, and therefore, they belong to the protein synthesis, aerobic-dependent pathway. Norfloxacin, a second-generation quinolone, is not lethal in the presence of chloramphenicol, but it is lethal under anaerobic conditions and, thus, belongs to the protein synthesis-dependent, aerobic-independent pathway. Ciprofloxacin, a second-generation quinolone, and other second- and third-generation quinolones, are lethal regardless of protein synthesis or aerobiosis, so they belong to the protein synthesis, aerobic-independent pathway [65,90].

In principle, gyrase-cleavage complexes could be removed by a protein (e.g., either a nuclease that cleaves next to the gyrase-DNA bond, a protease that processes the topoisomerase or a protein that specifically breaks the bond between the gyrase and the DNA) or by the dissociation of the gyrase subunits. For the first-generation quinolones that belong to the protein synthesis-dependent pathway, it is expected that said protein would be needed to be lethal. Whereas FQs like ciprofloxacin, which are lethal regardless of continued protein synthesis, might be lethal due to dissociation of the gyrase subunits. We will discuss these alternatives in the following sections.

#### 3.2.2. Nuclease and Protease Activity

Chen et al. [65] first suggested that there is a bacterial protein (or proteins) that can release the poisoned gyrase from the DNA. This suggestion was based on the observation that first-generation quinolones were not lethal if the synthesis of proteins was inhibited. This means that the lethality of first-generation quinolones depends on the presence of a protein that removes the gyrase from the DNA and that releases the lethal double-strand breaks, i.e., that protein would be responsible for killing by quinolones. Malik et al. [66] tried to find that protein by doing rounds of treatments with quinolones to find mutants that were bacteriostatic but not bacteriolytic. They were not able to map the mutations, and they suggested that multiple genes might be involved. To date, that protein(s) has not been found, although there are potential candidates that we will mention later.

In eukaryotes, several proteins are known to remove trapped topoisomerase-DNA complexes [91]. Notably TDP1 and TDP2, Tyrosyl-DNA-Phosphodiesterases 1 and 2, are well-characterised DNA repair proteins that remove the 3′ or 5′ tyrosyl-DNA adducts with eukaryotic topo I and topo II, respectively [92]. It is interesting to note that, in contrast to what has been hypothesised in bacteria (that there is a protein that can remove gyrase from the DNA and cause lethal breaks), in eukaryotes, this type of protein is involved in repair and not in killing [91,93,94,95]. This might be because the lethality of topoisomerase poisons in eukaryotes comes from DNA breaks that occur when replication forks collide. Whereas, with prokaryotes, the lethality mostly comes after the poisoned topoisomerases are removed from the DNA, releasing double-strand breaks, and, less importantly, when the replication forks are stalled. This means that if the poisoned topoisomerases are not processed in eukaryotes, they will be much more lethal than in prokaryotes. In any case, it is likely that the proteins that participate in the processing of poisoned gyrase are involved in both the killing and the repair, as the cleavage complexes need to be removed to be lethal and to be repaired.

Some evidence points to SbcCD and RuvAB as protein complexes that can remove poisoned topoisomerases in bacteria [81,96]. SbcCD is a nuclease complex that can make double-stranded breaks to release a protein attached to the DNA [97]. The deletion of *sbcCD* increased the sensitivity of cells to oxolinic acid but not to ciprofloxacin, and cells without *sbcCD* had more cleavage complexes in the presence of quinolones than wild-type cells, suggesting that SbcCD was needed to remove the gyrase [96]. However, it has not been tested directly (for example, by incubating purified SbcCD with DNA and gyrase) whether SbcCD can indeed cut the DNA near gyrase-bound DNA ends. The second protein complex, RuvAB, has helicase activity and was shown to remove topo IV-norfloxacin-DNA complexes [81]. Purified RuvAB was able to reverse cleavage complexes in vitro without the help of any other protein. How RuvAB causes the reversal of cleavage complexes is not known.

#### 3.2.3. Gyrase Subunit Dissociation

The second proposed mechanism to remove poisoned gyrase is the dissociation of gyrase subunits. When Chen et al. [65] found that second-generation quinolones, like ciprofloxacin, killed bacteria in the absence of protein synthesis, they hypothesised that it happened through the dissociation of gyrase subunits. As mentioned above, gyrase is a heterotetrametric enzyme (A_2_B_2_) formed of two GyrA and two GyrB subunits. If the GyrA subunits, somehow without the help of another protein, dissociate and leave the DNA, the lethal double-stranded breaks would be exposed. Malik et al. [66] tested this hypothesis by using a GyrA(A67S) mutant that presumably had an unstable GyrA-GyrA interface. They found that when they used nalidixic acid, cells were killed in the presence of chloramphenicol, and extracted nucleoids were fragmented (as happens with second-generation quinolones) [66]. This indicated that the mutation in GyrA caused a first-generation quinolone to stabilise the cleavage complex in a similar way to second-generation quinolones. However, even if gyrase subunits can dissociate because of the presence of a second-generation quinolone, their GyrA subunits would still be covalently bound to the DNA. Thus, further processing of GyrA would be needed to completely release it from the DNA.

#### 3.2.4. Chromosome Fragmentation

The final event of rapid quinolone killing is the fragmentation of the chromosome. Chromosome fragmentation has been observed in cells exposed to high concentrations of gatifloxacin, ciprofloxacin or oxolinic acid by measuring the sedimentation of the DNA [66] or by visualising the fragmentation of the nucleoids under the microscope [98]. As chromosome fragmentation is a hallmark of cell death [99], it is likely that it is a direct and quick cause of bacterial killing. 

#### 3.2.5. Reactive Oxygen Species (ROS) Formation

Apart from chromosome fragmentation, rapid quinolone killing has also been associated with the accumulation of intracellular reactive oxygen species (ROS). These are natural by-products of aerobic metabolism, and some examples include peroxide (H_2_O_2_), superoxide radical (O_2_^•−^) and hydroxyl radicals (^•^OH). Bactericidal antibiotics presumably increase ROS levels by disrupting the membrane, which causes the activation of the Krebs cycle [100]. In the Krebs cycle, reduced cofactors are formed. The reduced cofactors travel down the electron transport chain, where they can release electrons to oxygen molecules producing O_2_. Superoxide oxidises iron-sulphur clusters of respiratory dehydrogenases, which causes the release of iron. The iron, which is kept reduced by cellular reductants, then reduces H_2_O_2_ to ^•^OH, which can damage the DNA by oxidising DNA bases, creating aberrant base pairs, often leading to mutations [101]. When ROS accumulate in the cell, bacteria respond through the *oxyR*, *soxRS* and *rpoS* regulons. These regulons control the transcription of genes that degrade O_2_ (e.g., *sod* or superoxide dismutase) or that degrade H_2_O_2_ (e.g., *kat* or catalase) [102].

Several groups have shown a correlation between quinolone lethality and the accumulation of ROS [69,103,104]. They have done this by measuring levels of ROS and lethality after treating cells with quinolones and inhibitors of ROS or using strains overproducing or lacking enzymes that regulate oxidative stress (e.g., *sod* and *kat*). For example, Dwyer et al. [105] showed that, after norfloxacin treatment, ROS-related genes were upregulated, there was an increase in ^•^OH and no killing was observed when a ROS neutralizer was used. More recently, Hong et al. [69] found that all types of quinolones were lethal because of the accumulation of ROS.

Initially, there was some controversy about whether quinolones could indeed kill cells by the accumulation of ROS. Liu et al. [106] showed that quinolones did not increase the levels of ROS, and Keren et al. [107] found no correlation between ROS formation and quinolone lethality [106,107]. The disparities between these results and the ones from Dwyer et al. [105] were addressed in an exhaustive review [103], and since then, several studies have shown that ROS account, at least in part, for the lethality of quinolones [69,104,108]. Still, there are many unanswered questions around the ROS formation theory. For example, how do quinolones induce the formation of ROS? Norfloxacin treatment increases the production of metabolites of the Krebs cycle, which might trigger the formation of ROS [109], but the toxic effects of quinolones happen when they stabilise a cleavage complex, and the steps between the formation of quinolone-stabilised cleavage complexes and the formation of ROS have not been unravelled. Additionally, can quinolones induce the formation of enough ROS to kill the cell? The accumulation of intracellular ROS theoretically can only inhibit growth, and the mechanisms that cause cell death after ROS formation are not fully known [110,111]. Nevertheless, it is been shown that ROS can convert single-stranded DNA breaks into double-stranded breaks [112] and that the accumulation of ROS can be self-amplifying [104], so it is possible that ROS cause sufficient DNA damage to kill the cell.

### 3.3. Repair of Quinolone-Induced DNA Damage

Quinolones induce DNA damage that can kill the cells, but this damage can also be repaired [45]. The repair of quinolone-induced DNA damage was observed in quinolone-treated bacterial cultures that became denser after removing the drug [66]. This increase in density suggested that DNA breaks were resealed, as the longer the DNA is, the more viscous the solution is. Additionally, nucleoids of quinolone-treated cells became less fragmented after removing the quinolones [98], and the deletion of DNA repair proteins increased the susceptibility of bacteria to quinolones [113,114,115].

It is not clear how bacteria repair quinolone-induced damage. Due to the mechanism of quinolone killing and how eukaryotes repair poisoned topoisomerase-DNA cleavage complexes, it is possible that bacteria first have to remove the topoisomerase trapped on the DNA (for example, through the nuclease SbcCD or the helicase RuvAB). This would release the double-strand breaks and would activate stress responses such as the SOS response [51]. The SOS response is a cellular response to DNA damage that is controlled by the auto-repressor LexA and the activator RecA [116]. The SOS response is activated by all quinolones [88], but it is not the only stress response activated by quinolones. Pribis et al. [117] showed that, when exposed to sublethal ciprofloxacin, a subpopulation of cells undergoes SOS and the accumulation of ROS, which then activates the σ^S^ response. This is a general stress response that regulates the transcription of hundreds of genes, including genes involved in DNA repair. All these stress responses lead to the repair of DNA breaks through error-free (e.g., homologous recombination or nucleotide excision repair) or error-prone (e.g., translesion synthesis) DNA damage repair pathways [116,118]. Error-free repair pathways do not cause mutations, whereas the error-prone repair can generate mutations. This is one of the reasons why quinolones can trigger the appearance of mutations that cause quinolone antimicrobial resistance. 

## 4. Resistance to Quinolones

Antibacterial resistance towards the FQ drugs has arisen following its widespread use as a medication in both humans and animals [119,120,121]. In particular, during 2001–2006, the prevalence of FQ-resistant *E. coli* isolates in the UK increased from 6% to 20%. This then decreased slightly, to 17%, by 2010 [122]. Furthermore, even higher quinolone resistance rates of Enterobacteriaceae (such as *E. coli*) have been recorded across the globe; in 2015, it was reported that up to 30% of community-associated isolates from across the United States showed FQ nonsusceptibility [123]. Elsewhere resistance rates are very variable but can be as high as almost 100%, particularly in Asia [124,125]. The rising resistance towards FQs threatens their efficacy against a range of diseases, and scientific efforts have focused on understanding the mechanisms behind resistance and the different ways to combat the bacterial infections. The mechanisms include the upregulation of efflux pumps, a reduced ability to uptake the drug, plasmid-mediated resistance or actual mutations in the gyrase or topo IV genes (Figure 7) [49,61].

### 4.1. Mutations in DNA Gyrase and Topo IV

The mutations in gyrase and topo IV that confer resistance to quinolones are often found in a region termed the quinolone resistance-determining region (QRDR), which is between amino acids 67 and 106 in GyrA (*E. coli* numbering) or 63 and 102 in ParC [126]. There is also a QRDR found in GyrB between amino acids 426 and 447 and in ParE between amino acids 420 and 441, with the two most common mutations found to be D426N and L447E (*E. coli* numbering) [55,127,128,129]. However, the most prevalent quinolone-resistance mutations are found in GyrA. These mutations cluster near the active site tyrosines at the dimer interface [130]. Due to their specific interactions with the quinolone through the water-metal ion bridge, the residues most commonly mutated in ciprofloxacin-resistant strains are serine and aspartic acid/glutamic acid on helix IV in GyrA/ParC [5,7,49,61]. Resistance-conferring mutations outside the traditional QRDR have also been identified. For example, an A51V mutation results in a six-fold increase in ciprofloxacin resistance [131]. Furthermore, there have been reports of the decreased gene expression of topo IV in *Staphylococcus aureus*, increasing its MIC to premafloxacin (Figure 3) and ciprofloxacin by two–eight-fold. This was found to be caused by a point mutation in the promoter of the *grlB* (*parE*) gene, which reduced the expression of the gene, conferring an increase in the MIC [132].

### 4.2. Plasmid-Mediated Quinolone Resistance (PMQR) 

Early in the history of quinolones, it was reported that they were able to eliminate plasmids from bacteria [133,134,135], suggesting that they were unlikely to be subject to plasmid-mediated resistance. However, plasmid-mediated resistance to FQs has now been discovered. The first plasmid gene found to introduce bacterial protection against FQs was named *qnrA*, which was followed by the isolation of several related genes, including *qnrB* and *qnrS* [61,136]. Each gene codes for a different Qnr protein, and QnrA was the first of these to be characterised. QnrA was assigned to a family of proteins known as the pentapeptide repeat proteins (PRPs), due to their series of five amino acid tandem repeats throughout the total sequence of 218 amino acids [137]. Additional PRPs, including MfpA and McbG, were also shown to aid in FQ resistance [138,139,140]. MfpA was the first of the proteins to produce a successful crystal structure, and this revealed a 3D form that appeared to mimic the structure of DNA [141]. The beta-helix-like structure was also observed with other PRPs, such as Qnr proteins (e.g., AhQnr from *Aeromonas hydrophila* [142]). McbG was initially discovered to protect *E. coli* gyrase against Microcin B17, a natural antibacterial peptide toxin produced by Enterobacteriaceae. These bacteria produce McbG to defend their own gyrase during the production of the toxin [140,143]. The PRPs (and similar plasmid-encoded resistance proteins) likely evolved as defence mechanisms against natural threats, such as competing bacteria. The PRP structure suggests that their primary function is to mimic DNA when binding competitively to DNA-dependent enzymes, thus preventing the binding of inhibitors. Current research efforts are aimed at revealing molecular mechanisms of protection by PRPs, with current data indicating that subtle sequence variation can cause significant functional differences. For instance, not all PRPs can protect gyrase or topo IV against FQs [140]; conversely, Qnr cannot protect against Microcin B17 [140,144].

A second PMQR mechanism was revealed with the detection of the AAC(6′)-Ib-cr mutant protein. AAC(6′)-Ib-cr is an aminoglycoside 6’-N-acetyltransferase enzyme containing two point mutations, W102R and D179Y, that introduce the ability to acetylate (and so deactivate) some FQs [145]. The D179Y alteration is believed to aid in favourable pi-stacking interactions during enzyme-FQ binding, and the W102R mutation is thought to position the FQ, perhaps through hydrogen bonds with the C-3/C-4 oxygen atoms. Acetylation occurs at the amino nitrogen of the piperazine ring within second-generation FQs and may impact binding with the target enzyme [145].

### 4.3. Altered Drug Transport

Other identified chromosomal mutations that confer quinolone resistance include those involved with the uptake of the drug, the upregulation of efflux pumps and in the regulons that control the expression of these. In Gram-negative bacteria, modifications of the bacterial membrane either structurally by the reduction of the number of porins (via OmpA and OmpX) in the cell membrane or through the alteration of the porins themselves have been reported [7,146,147,148,149]. Additionally, the overexpression of various efflux pumps (also found in Gram-positive species) can lead to increased resistance [7,61,150,151]. Efflux describes the process by which bacteria are able to expel harmful compounds (such as antibiotics) using active transport proteins known as efflux pumps. Alterations to efflux genes can arise from both chromosomal mutations and via plasmids, which typically involve changes in regulatory proteins and de-repression of the efflux systems [5,151,152]. Efflux pumps span the membranes of both Gram-negative and Gram-positive species, and the overexpression of these proteins lowers the cytoplasmic concentration of drugs retained in the cell [153]. Efflux effects can cause low-level resistance alone but present an advantage for the evolutionary selection of high-resistance strains [154,155]. These efflux pumps can be classified into five families; those that are most relevant to FQ resistance are the major facilitator superfamily (MFS) in Gram-negative and -positive species and the resistance-nodulation-division superfamily (RND) in Gram-negative species [152,156]. 

Efflux pumps can have a range of substrate specificities. For example, FQ efflux systems tend to be broad-ranged and able to transport many drugs and toxic compounds. This means that, often, mutations in these efflux pumps can cause resistance to FQs and other drugs at the same time (cross-resistance) [153,157]. Many FQ-resistant strains carrying such efflux mutations are typically resistant to multiple drugs. Two examples of plasmid-based efflux mutants that induce FQ resistance are *oqxAB* and *qepA*, isolated from animal and clinical samples, respectively [158]. Many chromosomal efflux mutants have also been detected, including *norA*, *norB* and *norC*, within *St. aureus* strains. The corresponding pumps are multidrug transporters, though they do display some specificity towards the structure of FQ that they bind. NorA only transports the more hydrophilic FQs (such as norfloxacin and ciprofloxacin), whilst NorB and NorC transport norfloxacin, ciprofloxacin and the less hydrophilic compounds (such as moxifloxacin and levofloxacin) [159]. Interestingly, the overexpression of NorA not only causes low-level resistance, but it also increases the evolvability of ciprofloxacin resistance in *St. aureus* [154]. In *E. coli*, the overexpression of efflux pumps is often linked to mutations in MarRA, SoxRS and Rob regulons, which are involved in the regulation of these efflux pumps, as well as many other pathways in the cell [7,61,147,160,161,162,163].

## 5. Quinolone-Induced AMR

Along with the increase in FQ resistance that is seen with FQ use, there is also evidence that FQs may increase resistance to non-FQ antibiotics [164,165,166,167,168], particularly under sublethal or sub-MIC exposure. Treatments with sublethal FQs have been shown to increase mutation, recombination and persister formation rates, often leading to an increase in the frequency of resistance to non-quinolone antibiotics (Figure 7) [169,170,171,172,173,174,175].

### 5.1. Treatment with Quinolones Increases Resistance to Non-Quinolone Antibiotics (QIAR)

There is growing evidence, particularly from the livestock and veterinary sectors, that FQ use can lead to an increase in antibiotic-resistant isolates [176]. This resistance can be FQ resistance, resistance to non-FQ antimicrobials or multidrug resistance. Pereira et al. [166] looked at the resistance profiles of *E. coli* isolates from pre-weaned calves that were treated with enrofloxacin (Figure 3) or the cephalosporin ceftiofur for diarrhoea and respiratory diseases. They found that 77% of the isolates from the FQ-treated calves showed resistance to three or more antimicrobials, including ciprofloxacin, streptomycin, tetracycline, ampicillin, ceftiofur and chloramphenicol. In the study, only the calves treated with the FQ were significantly more likely to have non-susceptible *E. coli* isolates [166]. Similarly, a study on healthy chickens found an increase in the number of isolates resistant to doxycycline, amoxicillin and enrofloxacin in the commensal *E. coli* populations after treatment with enrofloxacin [164]. This was mirrored in more recent studies on commensal *E. coli* isolates from chickens and turkeys that were treated with enrofloxacin [167,168]. In the former study, multidrug resistance was identified in *E. coli* isolates from chickens treated with the FQ for *Salmonella* sp. infections [168], whilst, in the latter, *E. coli* isolates from turkeys treated with enrofloxacin were found to be resistant to ampicillin, despite ampicillin not being used in the study [167]. Further evidence comes from *Sa*. Typhimurium clinical isolates from pigs, which showed multidrug resistance following a single treatment of marbofloxacin (Figure 3) below the mutant prevention concentration [177]. This phenomenon is not peculiar to isolates from livestock; a strain of multidrug resistant *Sa. enteritidis* was isolated after a patient with a splenic abscess was treated with ciprofloxacin [178]. One of the first studies done in vitro looked at the resistance profiles of *Salmonella* spp. after repeated exposure to FQs and β-lactams. They found that mutants generated after treatments with various FQs showed reduced susceptibility to a wide range of the antibiotics tested (seven different antibiotics). This was in contrast to mutants generated under treatment with β-lactams, which showed reduced susceptibility to fewer antibiotics (five antibiotics) [179]. Similarly, it has been shown that the treatment of methicillin-resistant *St. aureus* with sublethal FQ further enhances methicillin resistance [169]. A similar effect is seen with *E. coli*, as cells become resistant to quinolone and non-quinolone antibiotics after exposure to sublethal FQ [180]. Alongside these studies that show increases in resistance to non-quinolone antibiotics after FQ treatment, there is also evidence that sublethal treatment with quinolones increases mutation rates, mutation frequencies and recombination [170,171,172,173,174,175].

### 5.2. Sublethal FQ Treatment Increases Mutation Rate

Quinolones are potent inducers of the SOS response [45,181,182,183]. As mentioned above, the SOS response is the bacterial response to DNA damage. It is regulated by RecA, a recombinase that is activated when there is DNA damage, and LexA, the repressor of the SOS regulon that autocleaves when RecA is active. The autocleavage of LexA results in the derepression of the SOS regulon that controls the expression of ~50 genes in *E. coli*, including three error-prone polymerases (PoI II, Pol IV and Pol V) that can introduce mutations [116]. Thus, it is not that surprising that quinolones could potentially increase the mutation rate. Indeed, many studies have shown that after the treatment with sublethal concentrations of FQs, there is an increase in mutation rate and mutation frequency [169,170,171,172,173,174,183,184,185,186,187]. 

Ysern et al. [183] were the first to show that the quinolones increase mutagenesis through the induction of the SOS response. They suggested that the mutagenic effect was through the upregulation of Pol V. In 2005, Gillespie et al. [174] demonstrated that the treatment of *Mycobacterium fortuitum* with sub-MIC ciprofloxacin was able to increase the mutation rate by 72–120-fold. A more systematic study of the role of SOS in the mutagenic effect of quinolones was performed by Cirz et al. [184]. Using a neutropenic murine thigh infection model, they found that, in pathogenic *E. coli*, LexA, the repressor of the SOS response, was required for the evolution of the resistance induced by treatment with ciprofloxacin. They concluded that the homologous recombination pathway was important in the repair of ciprofloxacin-induced DNA damage and that LexA cleavage was induced during the repair. This then caused the upregulation of the three error-prone polymerases, which, together, generated the mutations that conferred resistance [184]. The same group went on to show that this was also the case in *P. aeruginosa* [173] and *St. aureus* [172]. LexA, RecA and error-prone DNA polymerases were also found to be upregulated in *M. tuberculosis* when treated with sublethal doses of ciprofloxacin [175]. Some groups, however, have suggested that upregulation of the error-prone polymerases is not the only mutagenic pathway induced by treatments with sublethal quinolones. Song et al. [188] showed that, when all three of the error-prone polymerases were deleted, ciprofloxacin-induced deletions still occurred. Long et al. [189] suggested that the increased mutagenesis observed with norfloxacin was not only due to the error-prone polymerases but, also, by indirect effects of the antibiotic on the mismatch-repair system and DNA-oxidative repair mechanisms. Oxidative stress has also been suggested as the main mutagenic pathway of a range of antibiotics, including norfloxacin, by Kohanski et al. [165]. They demonstrated that treating *E. coli* MG1655 with sublethal concentrations of norfloxacin in vitro caused multidrug resistance, which was abolished by the use of the ROS scavenger thiourea.

### 5.3. Sublethal FQ Treatment Stimulates Recombination

Sub-MIC treatment with quinolones have also been shown to increase genetic recombination. Ciprofloxacin has been demonstrated to stimulate homologous recombination in *E. coli* [170]. This was shown to be RecA-dependent and only partially reliant on induction of the SOS response; an uncleavable LexA mutant reduced the recombination but did not abolish it [170]. Sublethal ciprofloxacin was also demonstrated to induce nonhomologous recombination. This recombination was surprisingly independent of the SOS response but did require the other recombination pathways, RecBCD or RecFOR [171]. This study also looked at the effect of sublethal ciprofloxacin on conjugative transfer. Although they found no significant increase in the transfer of a conjugative plasmid, they did see an increase in horizontal gene transfer of an antibiotic-resistance gene from the plasmid to the genome [171]. Along these lines, three different FQs where shown to stimulate generalised transduction in *Sa.* Typhimurium. This was demonstrated through the transfer of a kanamycin resistance gene from a multidrug-resistant strain to susceptible strains through a P22-like bacteriophage [190]. 

### 5.4. Sublethal FQ Stimulates the Formation of Persisters

Another factor is how sublethal FQs has been shown to increase the presence of persister cells in the population [113,191,192,193,194]. Persisters are cells that can survive lethal concentrations of antibiotics due to phenotypic (but not genetic) changes. How persisters manage to survive antibiotic action is not clear, though it has been suggested that they do it by inactivating the drug target or lowering the drug uptake [195]. Persister cells have been demonstrated to form as a result of induction of the SOS response [113,192,193] or through a cellular response to starvation [194]. This persister population has also been shown to allow long-term survival to the exposure to the quinolone by allowing mutations to accumulate within the population that are then selected for by the antibiotic [113,191]. 

How sublethal FQs induce mutation appears to be complex, much like the lethality of quinolones and the repair of quinolone-induced DNA damage. There appear to be several pathways that lead to an increase in mutation. However, mutation seems to be a consequence of repair, and this mutagenesis does not increase the evolvability of the bacteria [196]. More work is needed to understand the role FQs play in mutagenesis, as a better understanding of how these drugs induce mutation may enable strategies to be put in place that will reduce their role in the acquisition of antibiotic resistance.

## 6. Future Prospects

The significant rise in FQ resistance over the last 20 years has mirrored the poorly controlled usage of this broad-ranged drug class [197]. However, this period has also seen the discontinuation of many FQs due to serious adverse side effects, such as several third-generation analogues, including grepafloxacin and sparfloxacin. To tackle FQ resistance, sensible usage guidelines must be followed worldwide. There are some restrictions in place at present, like the prohibited prescription of ciprofloxacin for complicated UTIs, but controls must become more widespread to make a greater impact [198]. There have recently been restrictions placed on the use of FQs for mild bacterial infections, but this is due to potential side effects and not for antimicrobial resistance reasons [199]. Alongside this, the evidence that the misuse and sublethal exposure of FQs is potentially leading to an increase in mutagenesis and resistance to other antibiotics is of concern. A systematic review into the extent of the problem is underway [200].

The continued discovery of novel antibiotics that are potent against non-resistant and resistant strains of bacteria (especially those with MDR) is key in the fight against resistance. Though somewhat diminished, the development of FQs is still ongoing, with a few pharmaceutical companies currently conducting research on the synthesis of novel FQs. Delafloxacin (Figure 1) is an example of a FQ recently approved for clinical use [201]. The structure of delafloxacin has three distinct features: a 3-hydroxyazetidine ring substituent at C-7, a chlorine atom at C-8 and a bi-fluorinated aromatic ring at N-1. Overall, the structure is more acidic than other FQs, meaning the compound is more likely to be deprotonated (at its C-3 carboxyl group) at a neutral pH. It has been demonstrated to be effective against quinolone- and methicillin-resistant *St. aureus* due to its improved cellular uptake in acidic conditions [202,203]. Furthermore, delafloxacin shows dual targeting, meaning it inhibits both gyrase and topo IV with equal affinity, in contrast to older FQs (with the exception of those with C-8 methoxy groups). Dual targeting is believed to reduce the likelihood of drug resistance, as seen with moxifloxacin [204,205,206,207]. The exact explanation for this dual affinity is not necessarily clear. It was suggested that the C-8 methoxy substituent of moxifloxacin allows for dual targeting [208]; however, the lack of this group in delafloxacin suggests that it could not be caused by the methoxy group alone. However, it may simply be due to different FQs having differential affinities for gyrase and topo IV, depending upon their side groups, and the particular arrangement of amino acids in the binding pockets of the enzymes. Further crystal structures of the quinolone-enzyme-DNA complex should illuminate this.

The search for FQ alternatives is also incredibly important for combatting resistance. The screening of both natural and synthetic compounds against gyrase and topo IV provides the opportunity for the discovery of such inhibitors. For instance, an allosteric-binding pocket was discovered within gyrase when a thiophene-based compound (Figure 3) showed significant inhibition during high-throughput screens at GSK [209]. An even more potent analogue was developed from this compound, though it was found to be toxic in animal trials. Usefully, the thiophene compounds also stabilise the gyrase-DNA cleavage complex and do not show any cross-resistance with quinolones [209], suggesting that further investigation is warranted. Such discoveries provide the rationale for future research on compound designs and highlight the importance of screening. The determination of the structures of several gyrase and topo IV enzymes bound to antibiotic compounds raises the possibility of computational drug design methods being used in the search for new agents.

The quinazolinediones are a class of compounds that are structurally similar to FQs but lack the C-4 carboxyl substituent required for water-metal ion bridge formation [210] (Figure 3). It was hoped that these molecules may be unaffected by the two key target enzyme mutations and possess significant inhibition activity. Unfortunately, challenges have been found with their low potency and concurrent human topo II poisoning. This poisoning of the homologous human enzymes has indicated the quinazolinediones’ potential as anticancer agents, however [5]. The discovery of imidazopyrazinones (IPYs; Figure 3) as gyrase inhibitors that bind in a similar way to quinolones but have a different resistance profile are examples of other compounds that may hold promise as future antibiotics [211,212].

Another class of new topoisomerase inhibitor are the NBTIs (Novel Bacterial Topoisomerase Inhibitors). These compounds, which include the spiropyrimidinetriones and the triazaacenaphthylenes, bind adjacent to the quinolone-binding pocket and are not subject to the resistance mutations in the QRDR. They inhibit the enzyme by intercalating into the DNA at the dimer interface, stabilizing a pre-cleaved state, which increases the prevalence of single-stranded DNA breaks. Two of these NBTIs, gepotidacin and zoliflodacin, are in phase III clinical trials for the treatment of uncomplicated gonorrhoea (ClinicalTrials.gov Identifier: NCT04010539 and ClinicalTrials.gov Identifier: NCT03959527 respectively).

The understanding of low-level resistance mechanisms, such as PMQR, also provides helpful insight into the development of novel antibacterial drugs. For instance, the use of efflux pump inhibitors, in combination with FQs, has significant potential in the treatment of FQ-resistant species [213,214].

Although the induction of resistance by FQ treatment seems to be multifactorial, many groups have suggested that a combinatorial approach may be the way forward to reduce the mutagenic effects. Some have suggested the use of drugs that inhibit RecA or stop the cleavage of LexA [186,215]. This would reduce the mutagenesis but, with targeting RecA, also potentiate the quinolones themselves [186,216]. This has also been shown to potentially re-sensitise quinolone-resistant mutants [217]. Other groups have argued for the use for antioxidants agents such as N-acetylcysteine, which has been shown to reduce ROS and SOS induction without reducing the antimicrobial activity of ciprofloxacin [218].

As well as understanding resistance mechanisms, more work is needed to elucidate the exact path of quinolone lethality and repair. The exact mechanisms that lead to bacterial death, especially the role of ROS in lethality need to be found. The same is true of the mechanisms of repair; we still do not know what proteins remove quinolone-poisoned topoisomerases, despite these proteins having been known in eukaryotes for decades. Moreover, the role of the SOS response and other stress responses, like oxidative damage repair or general stress repair, need to be clarified. Additionally, different quinolones kill in different ways, and it is likely that their damage is repaired through different pathways. Answering all these fundamental questions will help us to better understand how quinolones work and what specific components of their lethality and repair pathways should be targeted in order to avoid the appearance of quinolone and non-quinolone resistance.

## Figures and Tables

**Figure 1 molecules-25-05662-f001:**
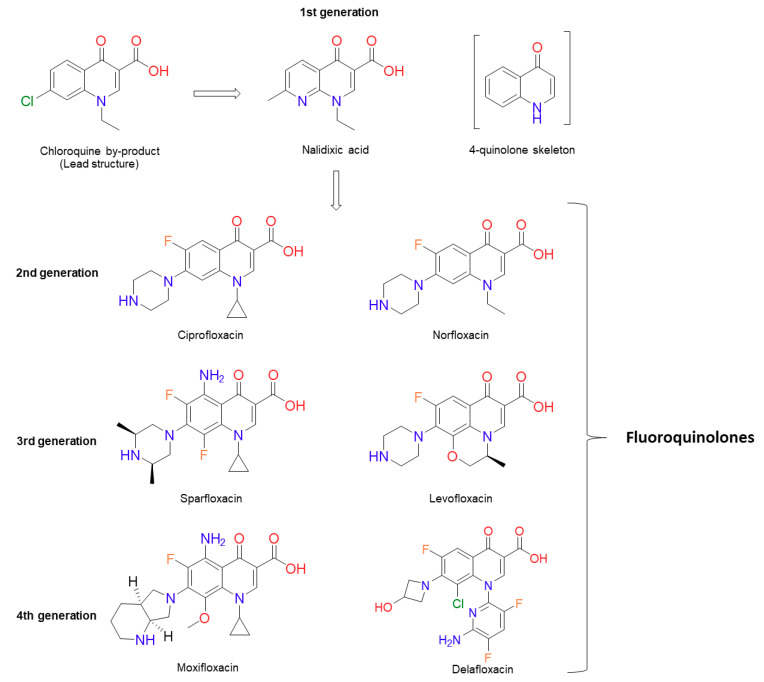
Chemical structures of several significant fluoroquinolones. Nalidixic acid (the first “quinolone”) is shown, along with the chloroquine by-product inspiring its synthesis. Note that nalidixic acid lacks the 4-quinolone core and instead contains a 1,8-naphthyridine nucleus.

**Figure 2 molecules-25-05662-f002:**
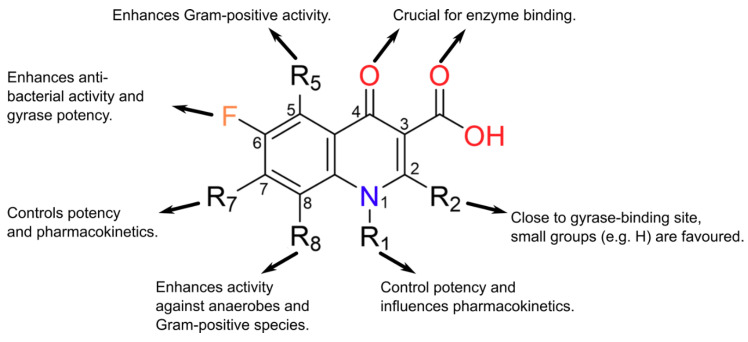
The observed structure-activity relationships (SARs) of quinolone core substitutions. Most often R5 = H, but sparfloxacin (a discontinued 3rd-generation fluoroquinolone (FQ)) has R5 = NH_2_ (diagram adapted from [32], with permission).

**Figure 3 molecules-25-05662-f003:**
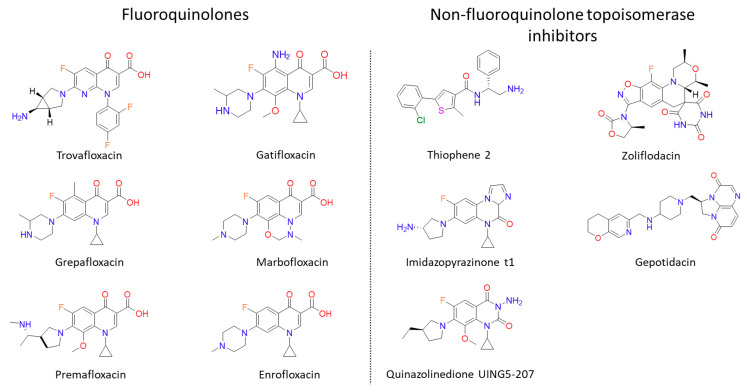
Structures of “new” fluoroquinolones and non-fluoroquinolone topoisomerase inhibitors discussed in the text. Gepotidacin and Zoliflodacin (Novel Bacterial Topoisomerase Inhibitors) are both in phase III clinical trials. Thiophene 2: *N*-(2-amino-1-phenylethyl)-5-(2-chlorophenyl)-2-methylthiophene-3-carboxamide, Imidazopyrazinone t1: 7-((3~[S])-3-azanylpyrrolidin-1-yl)-.5-cyclopropyl-8-fluoranyl-imidazo (1,2-a)quinoxalin-4-one and Quinazolinedione UING5-207: 3-Amino-1-cyclopropyl-7-((3R)-3-ethyl-1-pyrrolidinyl)-6-fluoro-8-methoxy-2,4(1H,3H)-quinazolinedione.

**Figure 4 molecules-25-05662-f004:**
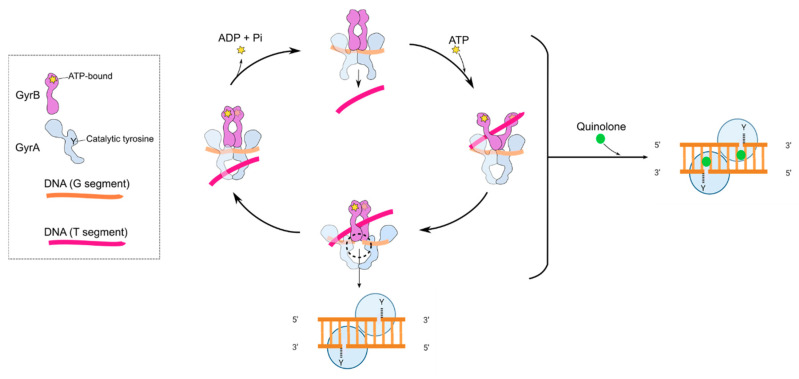
Cartoon showing the proposed mechanism of DNA supercoiling by DNA gyrase and how quinolones interfere with this mechanism by stabilising the cleavage complex. The inset shows the GyrA (blue) and GyrB (purple) subunits. Y indicates the position of the active site tyrosine, and the star indicates the position of the ATP-binding site. The G segment (orange) binds across the GyrA dimer interface. The GyrA C-terminal domain wraps the DNA (not shown) to present the T segment (pink) in a positive node. ATP binds to the N-terminal domain of GyrB, which closes the GyrB clamp (also known as the N-gate), capturing the T segment. The G segment is transiently cleaved, the GyrB domains rotate (not shown), the DNA gate widens and the T segment is transported through the cleaved G segment. The G segment is re-ligated, and the T segment exits through the GyrA C-gate. The hydrolysis of ATP and the leaving of ADP + Pi resets the enzyme for another cycle, although the exact timing of these reactions is unknown. The black-dashed circle and lower inset show the cleavage complex. The right-hand panel shows the binding of quinolones (green spheres) in the cleavage complex.

**Figure 5 molecules-25-05662-f005:**
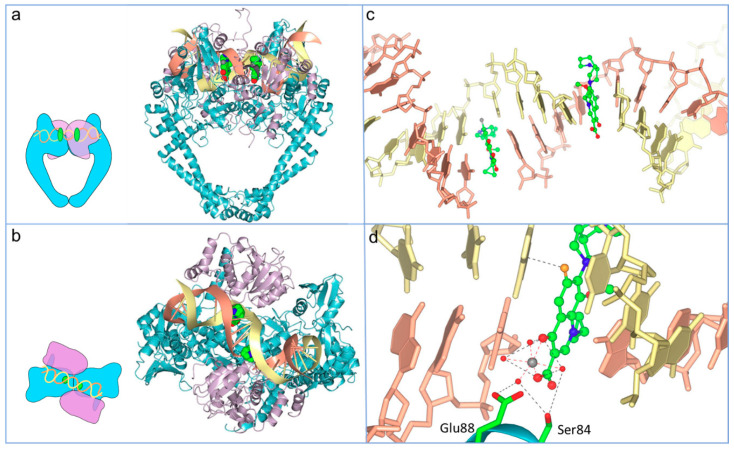
Overview of moxifloxacin binding to the topo IV-DNA complex (Protein Data Bank (PDB): 2XKK; [55]; figure modified with permission). (**a**) Front-faced view of moxifloxacin (Van der Waals model, green carbons) bound within the cleavage complex of *Acinetobacter baumannii* topo IV ParE (purple ribbons) N-terminal domain, fused to a ParC (blue ribbons) C-terminal domain and complexed with a 34-base pair (bp) heteroduplex DNA (yellow and coral ribbons). (**b**) View of the same complex from above. (**c**) Detail of moxifloxacin (ball and stick, green carbons) partially intercalated into the DNA bases at the break sites, spaced 4-bp apart. (**d**) Water-metal ion bridge formed between moxifloxacin, a noncatalytic Mg^2+^ (grey sphere), four water molecules (red spheres) and S84 and G88 of ParC.

**Figure 6 molecules-25-05662-f006:**
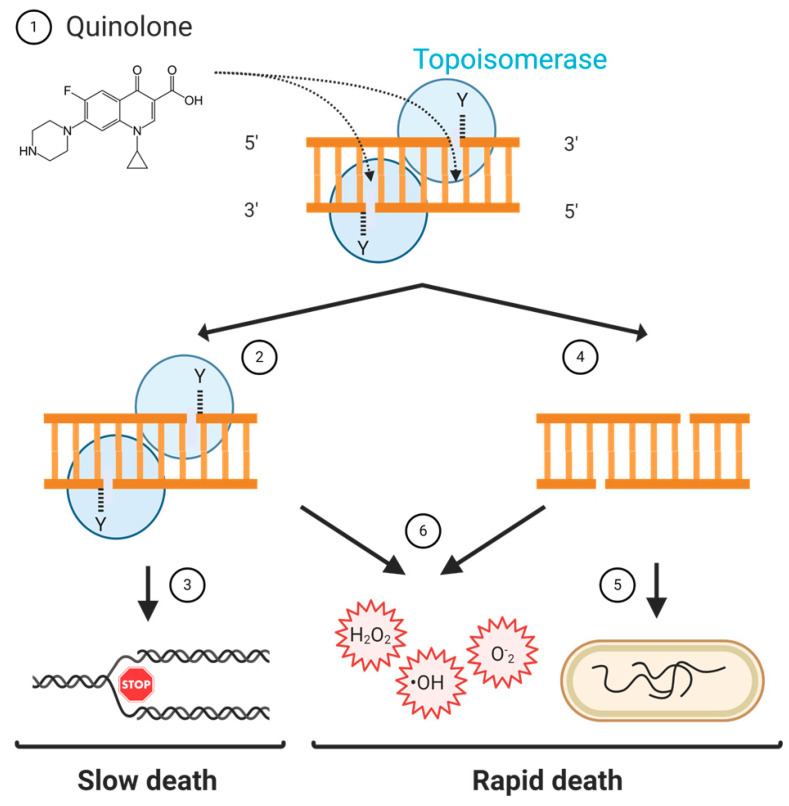
Model of quinolone lethality. (1) Quinolones stabilise the topoisomerase-DNA cleavage complex in which there is a double-strand break. (2) If the cleavage complex is not resolved, (3) replication and transcription cannot happen, which causes slow bacterial cell death. (4) If the topoisomerase is removed, the double-strand break is free, and if left unrepaired, (5) it leads to the fragmentation of the chromosome, which causes rapid bacterial cell death. (6) The stabilised cleavage complex, or the removal of the topoisomerase from the cleavage complex, might lead to the accumulation of reactive oxygen species (ROS) that can cause rapid bacterial cell death.

**Figure 7 molecules-25-05662-f007:**
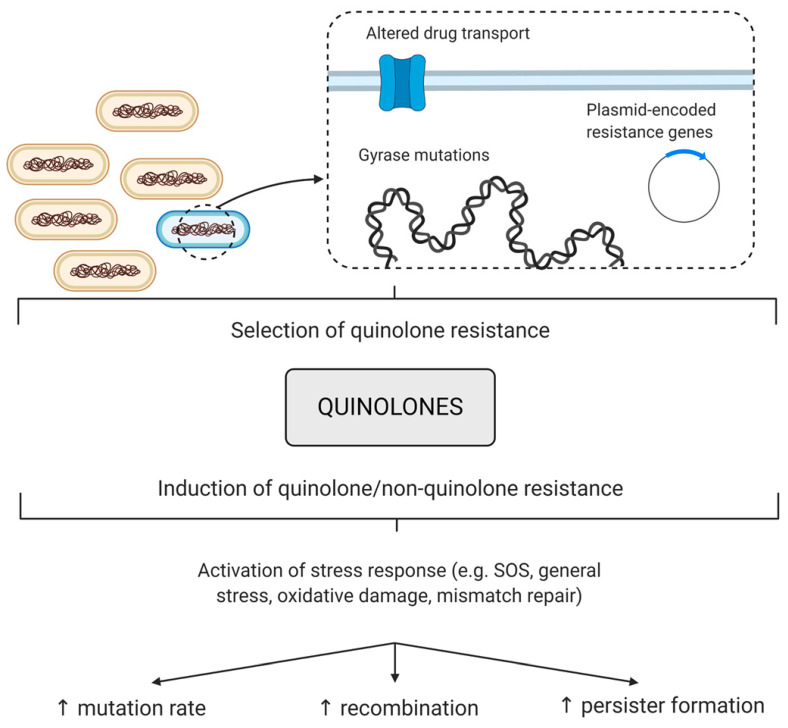
Contributions of quinolones to antibiotic resistance. Quinolones can select for quinolone resistance, which is caused by the upregulation of efflux pumps, mutations in DNA gyrase or DNA topoisomerase IV genes or plasmid-encoded resistance genes. Quinolones can also induce resistance to quinolones and non-quinolones antibiotics, presumably by the activation of a stress response that then increases the mutation, recombination or persister formation rates.

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
