# Peer review of "Quinolones: Mechanism, Lethality and Their Contributions to Antibiotic Resistance"

_molecules, 2020, doi:10.3390/molecules25235662_

Round 1

Reviewer 1 Report

This review is of interest and it includes a vast array of reports describing many aspects of quinolone research. While the authors made no attempt to review critically all the ROS-related results (the domain is such a mess, nobody wants to), this manuscript will be a nice contribution to the journal molecules provided the following point are addressed:

- A very relevant references has to be added somewhere in the intro: Mitscher, L. A. Bacterial topoisomerase inhibitors: quinolone and pyridone antibacterial agents. Chem. Rev. 2005, 105, 559-592.
- The conclusion is of interest but the depiction (along with the relevant references) of the other two classes of bacterial topoisomerase inhibitors which are acting via a similar mechanism of action (including the two in phase 3: Gepotidacine and Zoliflodacine) and appears to be devoid of cross resistance (due to the target mutation) would also be quite interesting especially since “thiophene 2” which no real potential is actually depicted.

Author Response

Response to reviewer’s comments for Manuscript: molecules-997058; Quinolones: Mechanism, Lethality and Their Contributions to Antibiotic Resistance. Authors: Natassja Bush, Isabel Diez-Santos, Lauren Abbott, Anthony Maxwell *

Firstly, we would like to thank both reviewers for taking the time to read and critically comment on our manuscript. We have made changes in the manuscript by Track Changes and we have included comments for every change. We shall address each reviewer’s comments and suggestions below:

Reviewer 1

This review is of interest and it includes a vast array of reports describing many aspects of quinolone research. While the authors made no attempt to review critically all the ROS-related results (the domain is such a mess, nobody wants to) (Thank you for raising this issue. We are aware of the controversies surrounding the antibiotic-induced-ROS killing and agree that a comprehensive and critical review of the subject is sorely needed, however, we felt that it was beyond the scope of this review), this manuscript will be a nice contribution to the journal molecules provided the following point are addressed:

- A very relevant references has to be added somewhere in the intro: Mitscher, L. A. Bacterial topoisomerase inhibitors: quinolone and pyridone antibacterial agents. Chem. Rev. 2005, 105, 559-592.

Thank you for drawing attention to this reference, it was omitted in error and we have now included it in line 77.

- The conclusion is of interest but the depiction (along with the relevant references) of the other two classes of bacterial topoisomerase inhibitors which are acting via a similar mechanism of action (including the two in phase 3: Gepotidacine and Zoliflodacine) and appears to be devoid of cross resistance (due to the target mutation) would also be quite interesting especially since “thiophene 2” which no real potential is actually depicted.

Thank you for this suggestion. We have included a short paragraph (lines 620 – 627) describing the potential of the NBTI’s and included the structures of Gepotidacin and Zoliflodacin in Figure 7.

Reviewer 2 Report

The work presented for review is an extensive review of the topic. Based on 216 papers (including 34 from 2017-2020), the authors presented the structure, classification, structure-activity relationship, as well as an overview of the mechanisms of action and resistance of quinolones. The subject is interesting and important due to the widespread use of fluoroquinolones and the marked increase in resistance to these compounds observed in recent years.

The  Authors should consider following suggestions:

  • Line 28: “tuberculosis (moxifloxacin) [3], prostatitis, community-acquired pneumonia, and skin and soft-tissue infections”- why the authors gave an example of a drug only in this one case, is it necessary?
  • Line 51: “nalidixic acid (although, technically, nalidixic acid is a 1,8 naphthyridone and not a true quinolone [1,2]),” - instead of "technically" it should be "structurally" or "in relation to the structure"
  • Line 250: “. 33.2.2. Nuclease/Protease” - Please correct
  • - Figure 7: the name of the compound "thiophene 2" is incorrect, it is a derivative of thiophene. The name thiophene is the name of a five-membered heteroaromatic ring containing a sulfur atom. This discrepancy also applies to other non-fluoroquinolone topoisomerase inhibitors presented in the table: imidazopyrazinon, quinazolinedione. A group of compounds that are derivatives of some basic structure is called, for example, quinolones or imidazopyrazinones. However, for a single compound it should be noted that it is derived from such a chemical structure.
  • the list of references requires many corrections: there are no abbreviations of the journal names.

Author Response

Response to reviewer’s comments for Manuscript: molecules-997058; Quinolones: Mechanism, Lethality and Their Contributions to Antibiotic Resistance. Authors: Natassja Bush, Isabel Diez-Santos, Lauren Abbott, Anthony Maxwell *

Firstly, we would like to thank both reviewers for taking the time to read and critically comment on our manuscript. We have made changes in the manuscript by Track Changes and we have included comments for every change. We shall address each reviewer’s comments and suggestions below:

Reviewer 2

The work presented for review is an extensive review of the topic. Based on 216 papers (including 34 from 2017-2020), the authors presented the structure, classification, structure-activity relationship, as well as an overview of the mechanisms of action and resistance of quinolones. The subject is interesting and important due to the widespread use of fluoroquinolones and the marked increase in resistance to these compounds observed in recent years.

The Authors should consider following suggestions:

Line 28: “tuberculosis (moxifloxacin) [3], prostatitis, community-acquired pneumonia, and skin and soft-tissue infections”- why the authors gave an example of a drug only in this one case, is it necessary?

Thank you for pointing this out, we have removed the moxifloxacin.

Line 51: “nalidixic acid (although, technically, nalidixic acid is a 1,8 naphthyridone and not a true quinolone [1,2]),” - instead of "technically" it should be "structurally" or "in relation to the structure"

You are correct, we have changed this to read “in relation to the structure”.

Line 250: “. 33.2.2. Nuclease/Protease” - Please correct. Done

- Figure 7: the name of the compound "thiophene 2" is incorrect, it is a derivative of thiophene. The name thiophene is the name of a five-membered heteroaromatic ring containing a sulfur atom. This discrepancy also applies to other non-fluoroquinolone topoisomerase inhibitors presented in the table: imidazopyrazinon, quinazolinedione. A group of compounds that are derivatives of some basic structure is called, for example, quinolones or imidazopyrazinones. However, for a single compound it should be noted that it is derived from such a chemical structure.

Thank you for pointing this out. We have included the full chemical names of the specific compounds in the figure legend (Figure 7)

the list of references requires many corrections: there are no abbreviations of the journal names.

Apologies, we have corrected these by updating Endnote and addressed any other abnormalities that were there.